# WHY NOT BOTH? COMBINING BELLMAN LOSSES IN DEEP REINFORCEMENT LEARNING

## ABSTRACT

Several deep reinforcement learning algorithms use a variant of fitted Q-evaluation for policy evaluation, alternating between estimating and regressing a target value function. In the linear function approximator case, Fitted Q-evaluation is related to the projected Bellman error. A known alternative to the projected Bellman error is the Bellman residual, but the latter is known to give worse results in practice for the linear case and was recently shown to perform equally poorly with neural networks. While insufficient on its own, we show in this paper that the Bellman residual can be a useful auxiliary loss for neural fitted Q-evaluation. In fact, we show that existing auxiliary losses based on modelling the environment's reward and transition function can be seen as a combination of the Bellman residual and the projected Bellman error. Experimentally, we show that adding a Bellman residual loss stabilizes policy evaluation, allowing significantly more aggressive target network update rates. When applied to Soft-Actor Critic—a strong baseline for continuous control tasks—we show that the target's faster update rates yield an improved sample efficiency on several Mujoco tasks, while without the Bellman residual auxiliary loss, fitted Q-evaluation would diverge in several such instances.

## 1 INTRODUCTION

One of the challenges of sequential decision making is that of credit assignment: to improve its policy, the learner needs to adequately determine the contribution of past actions to the overall performance of its behavior. In Reinforcement Learning (RL, Sutton & Barto 2018), credit assignment can be addressed by learning a Q function, which estimates future rewards of the agent for any state-action pair. Endowed with such a function, the problem of finding a policy maximizing the cumulative rewards simplifies to that of finding a policy greedily maximizing the Q function at the agent's state. A seminal deep RL algorithm following this general scheme is DQN (Mnih et al., 2015), which was able to learn a neural agent performing at human level on a range of Atari games and kickstarted a wave of impressive practical successes for deep RL.

DQN builds on the foundation of Fitted Q-Iteration (FQI) algorithms (Ernst et al., 2005; Riedmiller, 2005). FQI repeatedly i) computes a target function $\mathcal{T}Q_k$ by applying the Bellman operator $\mathcal{T}$ on the current Q function estimate $Q_k$, ii) computes a new Q function estimate $Q_{k+1}$ by minimizing the mean squared error to the target function $\mathcal{T}Q_k$. While other deep RL algorithms such as PPO (Schulman et al.), TD3 (Fujimoto et al., 2018) or SAC (Haarnoja et al., 2018) are a bigger departure from FQI than DQN, they still use a similar scheme for evaluating the Q-value of policies, which we denote more generally as Fitted Q-Evaluation (FQE). Due to its widespread usage in deep RL, understanding or improving FQE is thus an important research topic.

When the Q function is linear w.r.t. a fixed set of feature vectors, FQE shares similarities with LSTD (Bradtke & Barto, 1996), in that if FQE converges to a fixed point, this solution similarly minimizes the mean squared Projected Bellman Error (PBE). In the policy evaluation literature, an alternative to PBE is the mean squared Bellman Error (BE), also known as the Bellman residual (Schweitzer & Seidmann, 1985; Antos et al., 2006). There has been a long standing debate on whether one should minimize the PBE or the BE. For fixed feature sets, the PBE was generally considered better performing in practice (see e.g. discussion in Sec. 5.3. of Lagoudakis & Parr 2003), even though minimizing the BE can be more numerically stable (Scherrer, 2010). In the context of deep RL, Fujimoto et al. (2022) compared BE minimization to a neural FQE scheme and showed

that while the direct minimization of BE via gradient descent is effective at driving the BE close to zero (unlike FQE), the distance to the true value function is usually larger for BE minimization than that of FQE. We note that in the work of Fujimoto et al. (2022), both the environment and policy were deterministic, and thus the shortcomings of BE as a loss function were independent of the infamous double sampling issue which otherwise poses a practical challenge for BE minimization (Antos et al., 2006).

These negative results for BE minimization suggest that BE may not be a useful loss in deep RL. In this paper, we arrive at a similar conclusion to Fujimoto et al. (2022), in that BE minimization on its own does not perform as well as FQE[1]. The novel outcome of this work however, is that adding a BE auxiliary loss to FQE makes the latter more robust to higher update rates of FQE's target network—which is the number of gradient updates of the critic before the target network is replaced with the current network. Notably, we show that when the policy evaluation step of SAC (Haarnoja et al., 2018) is augmented with a BE term, increasing the update rate of the target network remains stable while improving sample efficiency. This is orthogonal to variants of SAC such as REDQ (Chen et al., 2020b) that improve sample efficiency by increasing the update-to-sample ratio—i.e. the number of gradient updates of the critic for every interaction with the environment. In this work it is the faster update rate of the target network that speeds-up the learning of Q functions, at the cost of an additional forward and backward pass on the critic incurred by the auxiliary BE term.

In an effort to explain the good performance of this new loss, we start by presenting the BE and PBE losses in Sec. 2, before proposing a way of combining them in Sec. 3. We show that this new loss is a lower bound to auxiliary losses previously studied in the literature based on modelling the agent's environment (Gelada et al., 2019; Chang et al., 2022). However, the exact loss introduced in Sec. 3 is expensive to compute and thus we propose a more frugal instantiation thereof in Sec. 4, that builds on FQE and simply adds a BE auxiliary term. This new policy evaluation scheme is evaluated on a set of on-policy policy evaluation problems before its integration to an actor-critic algorithm is assessed on several Mujoco locomotion tasks (Todorov et al., 2012) in Sec. 6.

## 2 Preliminaries

Let a Markov Decision Process (MDP) be defined by the tuple $(\mathcal{S}, \mathcal{A}, \mathcal{R}, P, p_0, \gamma)$, where $\mathcal{S}$ is a finite state space; $\mathcal{A}$ a finite action space; $\mathcal{R} : \mathcal{S} \times \mathcal{A} \mapsto [0, 1]$ a bounded reward function; $P : \mathcal{S} \times \mathcal{A} \mapsto \Delta(\mathcal{S})$ a transition function mapping a state-action pair $(s, a) \in \mathcal{S} \times \mathcal{A}$ to a distribution over $\mathcal{S}$ denoted $P(.|s, a)$; $p_0$ an initial state distribution and finally $\gamma$, a discount factor. We formalise our work with finite state and action spaces to streamline the exposition but the practical algorithm in Sec. 4 supports MDPs with continuous state-action spaces as demonstrated in our experiments. Given a stochastic policy $\pi : \mathcal{S} \mapsto \Delta(\mathcal{A})$, the Q function is defined for a pair $(s, a) \in \mathcal{S} \times \mathcal{A}$ by $Q^\pi(s, a) = \mathbb{E}\left[\sum_{t=0}^\infty \gamma^t R(s_t, a_t) \,\middle|\, s_0 = s, a_0 = a\right]$, for random variables $s_t$ and $a_t$, $t > 0$, where $s_t \sim P(.|s_{t-1}, a_{t-1})$ and $a_t \sim \pi(.|s_t)$. The V function is defined for $s \in \mathcal{S}$ as $V^\pi(s) = \mathbb{E}_{a \sim \pi(.|s)}[Q^\pi(s, a)]$ and the goal in RL is to find a policy $\pi^*$ such that $\pi^* = \arg\max_\pi \mathbb{E}_{s \sim p_0}[V^\pi(s)]$.

To do so, we focus in this paper on the class of (approximate) policy iteration algorithms (Sutton & Barto, 2018) that, given an initial policy $\pi_0$, alternate at every iteration $k$ between i) policy evaluation, to estimate the Q function $Q^{\pi_k}$ and ii) policy improvement, to find a new policy $\pi_{k+1}$ that selects actions with higher Q value than $\pi_k$. In the approximate setting, the policy improvement step can take several forms but since we will use off-the-shelves algorithms for this step we will not detail it further. Regarding the policy evaluation step, even though $\mathcal{S} \times \mathcal{A}$ is finite, we will assume that it is so large (e.g. if $\mathcal{S}$ is the space of all possible pixel images) that it warrants the usage of a function approximator to represent the Q function. Since we may not be able to learn $Q^\pi$ exactly, a first question of interest is which loss to use to measure the quality of an approximate Q function.

In the remainder of the paper, we will want to learn the Q function $Q^\pi$ of an arbitrary policy $\pi$. To approximate it, we use functions of the form $Q(s, a) = \phi(s, a) \cdot w$, where $\phi : \mathcal{S} \times \mathcal{A} \mapsto \mathbb{R}^K$ is a $K$ dimensional feature function and $w \in \mathbb{R}^K$ is a real-valued weight vector. Both the feature function and the weight vectors are learned—e.g. representing the hidden and final linear layers

---

[1]Although our conclusions are more nuanced than those of Fujimoto et al. (2022) since we have found a few datasets where BE minimization is competitive or outperforms FQE.

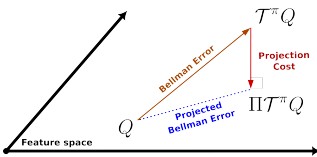

Figure 1: Illustration of the Bellman and projected Bellman losses for linear-in-feature Q functions.

of a neural network. Since the state-action space is finite, we define the $|\mathcal{S} \times \mathcal{A}| \times K$ matrix $\Phi$ as the concatenation of feature vectors for all state-action pairs, and we assume that $\Phi$ has full column rank (i.e. feature components are linearly independent) such that $\Phi^T\Phi$ is invertible. We let $\Pi = \Phi(\Phi^T\Phi)^{-1}\Phi^T$ be the orthogonal projection matrix w.r.t. the $L_2$ norm into the subspace spanned by $\Phi$, verifying for any vector $x \in \mathbb{R}^K$,

$$\Pi x = \arg\min_{y} \|x - \Phi y\|_2 . \tag{1}$$

We will overload notations of reward and Q functions to additionally designate matrices of size $|\mathcal{S} \times \mathcal{A}| \times 1$, and we denote by $P^\pi$ the $|\mathcal{S} \times \mathcal{A}| \times |\mathcal{S} \times \mathcal{A}|$ matrix such that entry with row index $(s, a)$ and column index $(s', a')$ of $P^\pi$ is given by $P^\pi_{(s,a),(s',a')} = P(s'|s,a)\pi(a'|s')$, i.e. the probability of being in $s'$ and executing $a'$ after having executed $a$ in $s$. Of course, $\Phi$, $\mathcal{R}$ and $P^\pi$ follow the same ordering of state-action pairs for their rows and columns. Finally, we define the Bellman operator $\mathcal{T}^\pi$ such that for any real matrix $Q$ of shape $|\mathcal{S} \times \mathcal{A}| \times 1$ we have $\mathcal{T}^\pi Q = \mathcal{R} + \gamma P^\pi Q$. A well known property of $\mathcal{T}^\pi$ is that it admits $Q^\pi$ as its unique fixed point, such that $\mathcal{T}^\pi Q^\pi = Q^\pi$.

### 2.1 POLICY EVALUATION LOSSES

Given an arbitrary function of the state-action space denoted by $Q$, we ideally want a loss function that measures some distance $\|Q - Q^\pi\|$ between this function and the true Q function. $Q^\pi$ being unknown, prior work in policy evaluation has resorted among other things to measuring how compliant is $Q$ w.r.t. the fixed point property of $\mathcal{T}^\pi$. Fig. 1 illustrates some of the main losses considered in the linear case, which we refer to as the case where $\Phi$ is fixed and only $w$ is learned. These losses are formally defined in the following way

**Bellman Error (BE).** BE is measured by the $L_2$ norm $\|Q - \mathcal{T}^\pi Q\|_2$. It is known that bounding the BE bounds the $L_\infty$ norm between $Q$ and $Q^\pi$ (Williams & Baird, 1994)

$$\|Q^\pi - Q\|_\infty \leq \frac{1}{1-\gamma} \|Q - \mathcal{T}^\pi Q\|_2 . \tag{2}$$

**Projected Bellman Error (PBE).** PBE adds an extra step by first projecting $\mathcal{T}^\pi$ into the span of $\Phi$, as illustrated in Fig. 1, before computing the distance with $Q$, giving the loss $\|Q - \Pi\mathcal{T}^\pi Q\|_2$. It is in general not possible to obtain similar guarantees on closeness to $Q^\pi$ to those of BE when minimizing the PBE (Scherrer, 2010), but it can behave in practice better than BE minimization and has led to popular algorithms in the linear case such as LSPI (Lagoudakis & Parr, 2003).

**Projection Cost (PC).** The PC refers to the cost of projecting $\mathcal{T}^\pi Q$ back into a linear function of the features and is measured by $\|\mathcal{T}^\pi Q - \Pi\mathcal{T}^\pi Q\|_2$. From the orthogonality of the projection, the above three quantities are related by the Pythagorean equation $\text{BE}^2 = \text{PBE}^2 + \text{PC}^2$. In the linear case PC is not optimized but is related to an inherent error due to a particular choice of a feature space (Munos, 2005). In deep RL, one can specifically optimize for feature spaces that have small PCs (Chang et al., 2022). This will be the starting point of our algorithm.

### 2.2 ALGORITHMS FOR POLICY EVALUATION

In the linear case, both BE and PBE admit a closed form solution (Lagoudakis & Parr, 2003). Of interest, the solution minimizing the PBE—as given by LSTD (Bradtke & Barto, 1996)—is

$$w = (\Phi^T(\Phi - \gamma P^\pi \Phi))^{-1}\Phi^T\mathcal{R}. \tag{3}$$

Fitted Q Evaluation (FQE, Ernst et al. 2005), described in Alg 1 can be summarized in the linear setting as iterating $Q_k = \Pi T^\pi Q_{k-1}$. If the algorithm reaches a point such that $Q_k = Q_{k-1}$, then $Q_k$

---

**Algorithm 1** Fitted Q Evaluation (FQE)

---

**Input:** Initial function $Q_0$, number of iterations $K$
**Output:** $Q_K$
 1: **for** $k = 1 \ldots K$ **do**
 2:     Compute target $\bar{Q}_k = T^\pi Q_{k-1}$
 3:     Find $Q_k = \arg\min_Q \left\| Q - \bar{Q}_k \right\|_2^2$
 4: **end for**

---

has the minimal PBE of zero. FQE can thus be seen as an iterative way of computing the solution of LSTD. Because of this indirect nature, it was shown in the realizable setting—when $Q^\pi$ is in the span of $\Phi$—that convergence of FQE is more restricted than the more direct computation of the LSTD solution (Perdomo et al., 2022). Despite this, FQE remains widely used because unlike LSTD, it can be straightforwardly extended to the deep RL setting, by replacing the orthogonal projection in Line 3 of the algorithm with a few gradient steps over the loss $\left\| Q - \bar{Q}_k \right\|_2^2$, evaluated on a minibatch of state-action pairs. In the remainder of the paper, when we talk of **update rate of the target network**, we mean the number of gradient descent steps performed on the loss $\left\| Q - \bar{Q}_k \right\|_2^2$, before moving to iteration $k + 1$ and computing a new target $\bar{Q}_{k+1}$ using the current Q-network $Q_k$.

BE minimization can also be performed by gradient descent in the deep RL case. The only added difficulty in this case is the double sampling issue which we discuss further in Sec. 4.1. The target computation in Line 2 is generally carried over a set of transitions $(s, a, r, s', a')$ instead of using the true Bellman operator. However, since we will follow this standard procedure when the model of the MDP is unknown, we keep the true Bellman operator notation for simplicity.

## 3 A NEW LOSS FOR POLICY EVALUATION

Our policy evaluation problem is to find a $Q = \Phi w$ close to $Q^\pi$. To measure closeness, PBE works well in practice for the linear case with algorithms such as LSTD, but for any feature function, as long as the LSTD solution exists, it will have a PBE of zero. As such, PBE on its own is not enough to discriminate the quality of features. To address this limitation, our starting idea is to use PBE for learning $w$ and combine it with one of the other two losses of Sec. 2.1.

Let a feature matrix $\Phi$ such that the LSTD solution exists, let $\bar{w}$ be this solution as given by Eq. 3. To decide on which loss to use for complementing the PBE, we note that for $\bar{w}$, the PBE is zero and from the Pythagorean equation, BE and PC become equal. Hence using any of these two losses is equivalent. Let the loss $\mathcal{L}$ be the Bellman error of $\bar{Q} = \Phi\bar{w}$, where $\bar{w}$ minimizes the PBE $\mathcal{L}(\Phi) = \left\| \bar{Q} - \mathcal{T}^\pi \bar{Q} \right\|_2$. $\mathcal{L}$ is only a function of $\Phi$, because $\bar{w}$ is implicitly defined from $\Phi$. To the best of our knowledge, no prior tried to minimize $L$ explicitly. However, this loss is related to several prior work adding auxiliary losses for modelling the MDP's reward and transition functions (Gelada et al., 2019; Chang et al., 2022).

### 3.1 RELATION TO PRIOR MDP MODELING LOSSES

To see the relation between $L$ and MDP modeling losses, let us first write the Q function $\bar{Q}$ through simple algebraic manipulations of Eq. 3, into a form that exhibits reward and transition models

$$\Phi\bar{w} = (I - \gamma\Pi P^\pi \Phi)^{-1} \Pi \mathcal{R}. \tag{4}$$

Eq. 4 is of course reminiscent of the expression of the true Q function $Q^\pi = (I - \gamma P^\pi)^{-1}\mathcal{R}$. The main difference is that we have exchanged i) the true reward function $\mathcal{R}$ with its projection $\bar{R} := \Pi\mathcal{R}$ into the span of $\Phi$ and ii) the true next state-action distribution $P^\pi$ with the projection of the expected next state-action feature $\bar{\Phi}' := \Pi P^\pi \Phi$ into the span of $\Phi$. Despite being perfectly model-free, the simple act of fixing a feature function $\Phi$ implicitly implies the choice of a model of the MDP $\bar{R}$ and $\bar{\Phi}'$, as discussed in Parr et al. (2008). Letting $\Phi' := P^\pi \Phi$ we can then relate $\mathcal{L}$ with the best linear reward model $m_r = \arg\min_m \left\| \mathcal{R} - \Phi m \right\|_2^2$ and best transition model $M_\Phi = \arg\min_M \left\| \Phi' - \Phi M \right\|_F^2$ of the MDP, where $\|A\|_F = \sqrt{\sum_i \sum_j a_{i,j}^2}$ is the Frobenius norm.

---

**Algorithm 2** Neural FQE gradient (+ DouBel + Bias correction)

---

**Input:** Neural function $Q_{\theta,w} = \phi_\theta \cdot w$, target network $Q_{\text{targ}}$, on policy transition sample $(s, a, r, s', a')$, $g_\beta : \mathcal{S} \times \mathcal{A} \mapsto \mathbb{R}$

**Output:** Gradients $\nabla_\theta$, $\nabla_w$ and $\nabla_\beta$

1: Compute target: $T = r + \gamma Q_{\text{targ}}(s', a')$
2: Feature function gradient: $\nabla_\theta (Q_{\theta,w}(s,a) - T)^2$ $+ \lambda \nabla_\theta (r + \gamma Q_{\theta,w}(s', a') - Q_{\theta,w}(s, a))^2$
  $- \lambda \gamma^2 \nabla_\theta (g_\beta(s, a) - Q_{\theta,w}(s', a'))^2$
3: Linear layer gradient: $\nabla_w (Q_{\theta,w}(s, a) - T)^2$
4: Gradient for $g$: $\nabla_\beta (g_\beta(s, a) - Q_{\theta,w}(s', a'))^2$

---

**Proposition 3.1.** *Let $\Phi$ be a feature matrix such that the LSTD solution $\bar{w}$ exists. Then*

$$L(\Phi) \leq \|\mathcal{R} - \Phi m_r\|_2 + \lambda_{\bar{w}} \|\Phi' - \Phi M_\Phi\|_F,$$

*with $\lambda_{\bar{w}} := \gamma \|\bar{w}\|_2$.*

*Proof.*

$$\mathcal{L}(\Phi) = \|\mathcal{T}^\pi \bar{Q} - \Phi \bar{w}\|_2, \tag{5}$$

$$= \|\mathcal{R} + \gamma \Phi' \bar{w} - (\bar{R} + \gamma \bar{\Phi}' \bar{w})\|_2, \tag{6}$$

$$\leq \|\mathcal{R} - \bar{R}\|_2 + \|\gamma(\Phi' - \bar{\Phi}')\bar{w}\|_2, \tag{7}$$

$$\leq \|\mathcal{R} - \bar{R}\|_2 + \lambda_{\bar{w}} \|\Phi' - \bar{\Phi}'\|_F, \tag{8}$$

$$= \min_{m_r, M_\Phi} \|\mathcal{R} - \Phi m_r\|_2 + \lambda_{\bar{w}} \|\Phi' - \Phi M_\Phi\|_F. \tag{9}$$

Eq. 6 uses the form of $\bar{Q}$ from Eq. 4. Inq. 7 and Inq. 8 respectively use the triangle and Cauchy-Schwarz inequalities. Finally, the last equality is due to the property of the projection given in Eq. 1, that is $\bar{R}$ and $\bar{\Phi}'$ are the best linear approximations of $\mathcal{R}$ and $\Phi'$ in the $L_2$ sense.

The interpretation of Eq. 9 is that the loss $\mathcal{L}(\Phi)$ can be upper bounded by two terms, one measuring how well can a linear model of $\Phi$ approximate the reward function, and the second term relates to the "self-predictiveness" of $\Phi$, i.e. how well can a linear model of $\Phi$ predict the expected next state-action features $\Phi'$, with $\lambda_{\bar{w}}$ making the trade-off between these two terms, depending on the scale of $\bar{w}$. $\mathcal{L}$ being a Bellman error, we know from Eq. 2 that $\mathcal{L}$ will upper bound the distance to the true Q function. Thus, minimizing Eq. 9 minimizes the distance to the true Q function, which justifies deep RL algorithms that add auxiliary losses for learning accurate models of the MDP (Gelada et al., 2019; Chang et al., 2022), even-though they are model-free and the models are not used. However, as we are not using these models, the research question that naturally arises is *can we directly use the Bellman error $\mathcal{L}$ as an auxiliary loss instead of its upper bounding model learning terms in Eq. 9?*

## 4 PRACTICAL IMPLEMENTATION

We now derive a practical algorithm for learning $Q$ following insights of Sec. 3. Unfortunately, optimizing $\mathcal{L}$ via gradient descent is challenging because $\bar{w}$ needs to be recomputed after each gradient step and LSTD itself can be numerically unstable (Scherrer, 2010). Instead, we will use $\mathcal{L}$ as an auxiliary loss to a standard neural FQE procedure and use the weight $w$ learned by FQE as an approximation to $\bar{w}$ for the current feature function. Algorithmically, this will be very close to the deep MDP procedure (Gelada et al., 2019), with the difference that we swap the model learning auxiliary losses—as they appear in Eq. 9—with the Bellman error in Eq. 5.

This augmented FQE procedure will involve two Bellman errors, and we will refer to it as the Double Bellman (DouBel) loss. Alg. 2 shows the gradient computation of neural FQE, and highlights in blue the contribution of the DouBel loss. Importantly, in Line 2, in the blue term the gradient w.r.t. the feature function flows through both the Q function computed at $(s, a)$ *and* $(s', a')$. However, the additional Bellman error has no contribution to the gradient of the linear part in Line 3. Indeed, the additional term in the DouBel loss aims at reducing the projection cost at $w$ but not for learning $w$, which is carried out by the standard FQE procedure.

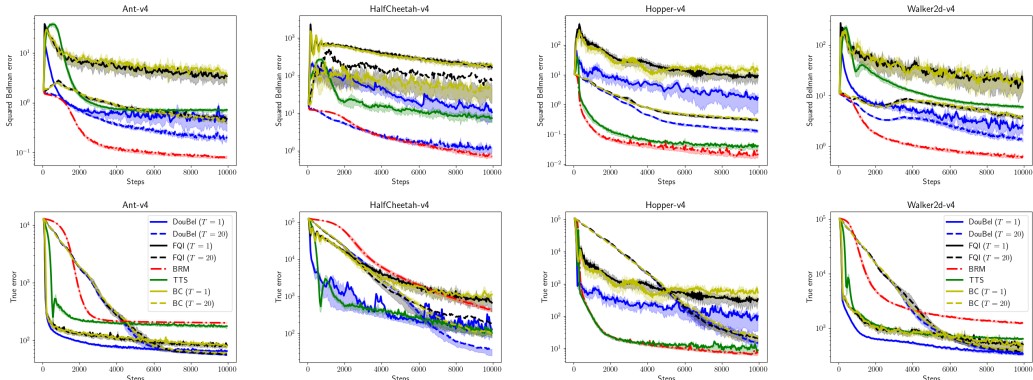

Figure 2: Bellman error (top) and corresponding distance to the true Q function on 4 datasets, showing the mean as well as the 25% and 75% quantiles. Results averaged over 20 runs.

### 4.1 DEALING WITH THE GRADIENT BIAS

In the blue error term of Line 2, Alg. 2, we have replaced the true Bellman operator $\mathcal{T}^\pi$ with a sample estimate using a single transition, i.e. we are estimating $(\mathcal{R}(s,a) + \gamma\mathbb{E}_{s',a'}Q(s',a') - Q(s,a))^2$ with a single sample for the expectation inside the square. Unfortunately, this is a biased estimator of the squared Bellman error with bias $\gamma^2\mathrm{Var}[Q(s',a')]$ (Antos et al., 2006), and is wildly referred to as the double sampling issue. To deal with the bias, a first approach is to simply ignore it. Indeed, many environments have little to no noise, and algorithms such as SAC quickly reduce the entropy of the policy making $\gamma^2\mathrm{Var}[Q(s',a')]$, and hence the bias, small.

The second approach is to correct the bias by estimating and substracting the variance term, as previously shown in Chang et al. (2022); Chen & Jiang (2019); Antos et al. (2006). Let $g : \mathcal{S} \times \mathcal{A} \mapsto \mathbb{R}$ be a function such that $g = \arg\min_f \|f - P^\pi Q\|_2$, where the minimization is over all possible real functions of the state-action space. In this case, we have that

$$g(s,a) = \mathbb{E}[Q(s',a') \mid s' \sim P(.|s,a), a' \sim \pi(.|s')], \tag{10}$$

and if for a sample transition $(s,a,r,s',a')$ we substract $\gamma^2(g(s,a) - Q_{\theta,w}(s',a'))^2$, we will be substracting $\gamma^2\mathrm{Var}[Q(s',a')]$ in expectation of $s'$ and $a'$, eliminating the bias. This can be simplified when the environment is deterministic by using a function of $s'$ instead of $s$ and $a$ to learn the expected next state-action Q function. Then, $g$ becomes $g(s') = \mathbb{E}[Q(s',a') \mid a' \sim \pi(.|s')]$ which is simply the value function. We use this latter form in our experiments.

The red highlights in Alg. 2 show the additional changes for correcting the bias of the squared Bellman error. In practice, the assumption that $g$ will perfectly approximate $\mathbb{E}[Q(s',a')]$ is not realistic and instead the bias correction will introduce an error term $(g(s,a) - \tilde{g}(s,a))^2$ of its own due to using an approximation $\tilde{g}$ of $g$. Hence, we will consider both approaches outlined in this section for dealing with the bias in the experiment section, in case $(g(s,a) - \tilde{g}(s,a))^2$ is higher than $\gamma^2\mathrm{Var}[Q(s',a')]$.

## 5 RELATED WORK

In supervised learning, finding rich feature functions for large input spaces remains an active area of research (e.g. Chen et al. 2020a; Jaiswal et al. 2020). The added challenge in RL is that finding a good representation is a continual process as new data is collected (Dabney et al., 2021). Even when the feature function is fixed, there has been a long debate on whether to use the Bellman or the projected Bellman error for learning the linear part. However, in the current RL research landscape, the Bellman error is seldom optimized and in fact several negative results have discouraged its use (Fujimoto et al., 2022; Geist et al., 2017), especially in the off-policy setting. On the other hand, FQE became the prominent policy evaluation scheme in deep RL and recent work has focused on adding auxiliary losses to FQE instead of completely changing the learning procedure: algorithms such as DeepMDP (Gelada et al., 2019) or SPR (Schwarzer et al., 2020) aim at learning features that

Table 1: Distance to the true Q function—mean and 95% CI—on 12 datasets. Numbers next to algorithm names indicate the number of gradient steps before the target network is updated.

|  | | DouBel (1) | DouBel (20) | FQE (1) | FQE (20) | BRM | TTS | BC (1) | BC (20) |
|---|---|---|---|---|---|---|---|---|---|
| | Ant_0 | $252.5 \pm 7$ | $\mathbf{229.3 \pm 6}$ | $314.1 \pm 30$ | $235.4 \pm 6$ | $256.8 \pm 1$ | $263.1 \pm 7$ | $325.2 \pm 26$ | $235.4 \pm 5$ |
| | Ant_1 | $545.0 \pm 22$ | $\mathbf{484.8 \pm 8}$ | $637.8 \pm 31$ | $494.0 \pm 9$ | $546.5 \pm 1$ | $544.2 \pm 10$ | $776.0 \pm 75$ | $503.2 \pm 15$ |
| | Ant_2 | $66.1 \pm 5$ | $\mathbf{55.4 \pm 1}$ | $80.6 \pm 4$ | $57.4 \pm 2$ | $200.5 \pm 1$ | $174.4 \pm 7$ | $83.7 \pm 3$ | $59.6 \pm 2$ |
| | Hop_0 | $126.0 \pm 71$ | $14.3 \pm 2$ | $269.2 \pm 69$ | $19.7 \pm 4$ | $\mathbf{6.4 \pm 1}$ | $8.8 \pm 2$ | $570.0 \pm 97$ | $22.0 \pm 7$ |
| | Hop_1 | $272.3 \pm 56$ | $\mathbf{95.4 \pm 6}$ | $723.4 \pm 158$ | $107.9 \pm 7$ | $201.2 \pm 12$ | $126.3 \pm 8$ | $822.1 \pm 204$ | $99.0 \pm 4$ |
| Datasets | Hop_2 | $634.5 \pm 166$ | $\mathbf{128.1 \pm 9}$ | $1658.1 \pm 245$ | $175.2 \pm 18$ | $344.3 \pm 20$ | $247.5 \pm 8$ | $1297.7 \pm 241$ | $147.2 \pm 8$ |
| | Walk_0 | $\mathbf{241.5 \pm 8}$ | $265.2 \pm 11$ | $450.2 \pm 106$ | $298.2 \pm 7$ | $1836.2 \pm 39$ | $629.8 \pm 19$ | $478.8 \pm 98$ | $310.6 \pm 14$ |
| | Walk_1 | $\mathbf{339.9 \pm 15}$ | $357.8 \pm 12$ | $493.3 \pm 55$ | $380.3 \pm 26$ | $1240.4 \pm 37$ | $634.0 \pm 14$ | $496.2 \pm 56$ | $398.3 \pm 21$ |
| | Walk_2 | $251.1 \pm 22$ | $\mathbf{220.5 \pm 6}$ | $412.6 \pm 47$ | $238.2 \pm 16$ | $475.5 \pm 13$ | $280.5 \pm 4$ | $360.9 \pm 39$ | $237.3 \pm 17$ |
| | HalfC_0 | $\mathbf{17.6 \pm 2}$ | $17.7 \pm 1$ | $29.8 \pm 5$ | $23.6 \pm 2$ | $23.0 \pm 1$ | $19.2 \pm 1$ | $51.1 \pm 13$ | $22.0 \pm 1$ |
| | HalfC_1 | $141.3 \pm 45$ | $\mathbf{34.9 \pm 7}$ | $788.5 \pm 242$ | $189.6 \pm 122$ | $420.8 \pm 32$ | $119.1 \pm 24$ | $1122.9 \pm 198$ | $101.9 \pm 9$ |
| | HalfC_2 | $\mathbf{684.8 \pm 15}$ | $690.5 \pm 16$ | $721.2 \pm 25$ | $724.0 \pm 35$ | $2771.6 \pm 24$ | $2089.2 \pm 54$ | $720.4 \pm 14$ | $723.3 \pm 29$ |

can be "self-predictive" in the sense that $\phi(s, a)$ can predict next state-action feature $\phi(s', a')$. This is similar to the work of Chang et al. (2022) or Song et al. (2016) that aim at learning features with low inherent Bellman error (Munos, 2005), at the exception that the 'self-prediction' is limited to linear models in the latter two papers.

In contrast, in our work minimizing the Bellman error ensures that the projection cost is low for a $w$ of interest, but not necessarily low for all possible vectors $w$. This is in a sense a way of applying the value equivalence principle (Grimm et al., 2020) from the model-based RL literature to prior work model-free RL algorithms that model the MDP through auxiliary losses: we do not need features that capture all of the information about future features but only information useful for predicting future rewards.

Although derived from different principles, the work most similar to ours algorithmically is that of Chung et al. (2018), where a slow learner minimizes the squared TD error (Sutton & Barto, 2018) to learn the feature function while a fast learner uses linear TD algorithms such as TDC (Sutton et al., 2009) to learn the linear layer $w$. The main difference with our work is that the squared TD error in prior work is computed using another weight vector $w'$ which loses the insights of Sec. 3, that we want minimal projection cost for the current $w$, and the relation this has with the distance to the true Q function. Comparisons with this work is included in the next section.

## 6 EXPERIMENTS

To evaluate the proposed policy evaluation loss, we conduct two sets of experiments. The first set (Sec. 6.1) is in a controlled setting where the exact Q function of the policy is known. In this experiment the DouBel loss is compared to various policy evaluation baselines from the literature. In the second set of experiments (Sec. 6.2), we investigate whether integrating the DouBel loss to an existing state-of-the-art deep RL algorithm can yield any practical gains.

### 6.1 POLICY EVALUATION

We run PPO (Schulman et al.) for one million time-steps on 4 Mujoco (Todorov et al., 2012) environments and every 10K steps, we collect rollouts from the deterministic counterpart of the current policy. Each dataset contains 5000 transitions with trajectories of length at most 1000 steps. If after 1000 steps the MDP is not at a terminal state, we roll out an additional 2000 steps that are not stored but serve the computation of the true Q function. Because the environments and policies are deterministic, we can compute an extremely accurate estimation of the true Q function for each state along a trajectory using this single trajectory. For each environment, we select 3 datasets where the undiscounted return is around 1000, around 2000 and at the highest observed return. On each dataset we run a policy evaluation algorithm on a neural approximator with three hidden layers, ReLU activations and 256 neurons on each layer. The considered baselines are i) Fitted-Q Evaluation (vanilla version of Alg. 2) with label **FQE** ii) FQE with added DouBel loss (Alg. 2 plus the blue term, using $\lambda = 1$) without bias correction since both environment and policy are deterministic, labeled **DouBel** iii) Bellman residual minimization which performs gradient descent on $\|Q_{\theta,w} - \mathcal{T}^\pi Q_{\theta,w}\|_2^2$, with label **BRM** iv) Two timescale networks (Chung et al., 2018), using

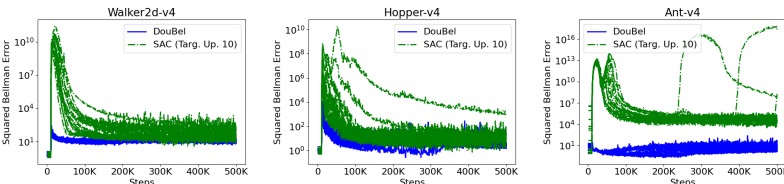

Figure 3: Episodic return for DouBel and SAC with three hidden layers for the critic, except the black curve that has 2 hidden layers. Results averaged over 20 runs showing mean and 95% CI.

Figure 4: Squared Bellman error of DouBel and SAC (T10) for each of the 20 runs. Both algorithms use a two hidden layer critic in this experiment.

linear TD as a fast learner with 10 times the learning rate of the inner layers, labeled **TTS** and finally v) FQE with auxiliary reward and next state feature approximation. The next state feature prediction is that of a target network as done in Chang et al. (2022), with label **BC**.

We additionally experiment with two hyper-parameter values for the update rate of the target network: updating it every 20 gradient steps or every gradient step—equivalent to not having a target network. This applies to FQE, DouBel and BC, whereas BRM and TTS do not have a target network and behave as the latter update regime. During learning (10K gradient steps), we record every 50 gradient steps the squared Bellman error and the squared distance to the true Q function on the 5000 transitions. All results are averaged over 20 independent runs, and Table 1 shows the mean distance to the true Q function and its 95% Confidence Interval (CI) on all 12 datasets, while Fig. 2 shows both the Bellman and true error as a function of time on a selection of 4 datasets—with the rest deferred to the appendix.

The top row of Fig. 2 shows the BE and without surprise BRM is the best at minimizing this loss. However, as discussed in Fujimoto et al. (2022), this does not necessarily translate in terms of true error on the bottom row, except for a few instances. Table 1 shows that BRM performs best on environments with lower dimensional states (Hopper and HalfCheetah) and when returns are not too high. When the update rate of FQE is of 20, FQE(20), DouBel(20) and BC(20) follow the same general tendency in terms of distance to the true error, but DouBel(20) consistently outperforms the other two on all 12 datasets, whereas BC(20) and FQE(20) are closer to each other, corroborating the results of Chang et al. (2022). The biggest difference between FQE and DouBel manifests on the faster update rate of the target, as the additional BE term allows a more effective reduction of the Bellman and the true errors. In contrast, BC(1) does not have the same effect on FQE. The only other baseline that rivals DouBel(1) in learning speed is TTS, which even outperforms it on a few datasets but completely misses the marks on others, typically when BRM similarly fails. Overall, when the update rate of the target network is slow, we found FQE to be a competitive algorithm. But the slow update rate has a noticeable impact on learning speed, sometimes requiring orders of magnitude more gradient steps to reach the same performance of the faster algorithms. As such, it is not surprising that increasing the number of gradient updates can have a noticeable effect on the sample efficiency of deep RL algorithms (Chen et al., 2020b) because FQE is slow. However, since DouBel loss minimization seems more robust to faster target update rates, we will investigate whether it can constitute an alternative way of improving sample efficiency of deep RL.

## 6.2 INTEGRATION TO SOFT-ACTOR CRITIC

In this second set of experiments, we integrate the DouBel loss into the `stable-baseline3` (Raffin et al., 2021) version of SAC (Haarnoja et al., 2018). Because the policy is now stochastic, the

Table 2: Undiscounted returns after 500K training steps on Mujoco environments. Results averaged over 20 runs showing mean and $95\%$ CI.

| | DouBel | DouBel (NB) | SAC | SAC (T10) | DouBel (H3) | DouBel (NB, H3) | SAC (H3) |
|---|---|---|---|---|---|---|---|
| **Walker2d** | $3684 \pm 160$ | $3606 \pm 122$ | $2608 \pm 449$ | $381 \pm 191$ | $\mathbf{4275 \pm 179}$ | $3928 \pm 151$ | $3478 \pm 490$ |
| **Ant** | $2978 \pm 408$ | $3613 \pm 435$ | $3323 \pm 334$ | $-198 \pm 208$ | $\mathbf{3801 \pm 582}$ | $3717 \pm 623$ | $3290 \pm 525$ |
| **Hopper** | $2387 \pm 300$ | $2356 \pm 283$ | $2778 \pm 248$ | $1676 \pm 400$ | $2844 \pm 243$ | $\mathbf{3077 \pm 212}$ | $2864 \pm 186$ |
| **Humanoid** | $4947 \pm 71$ | $5005 \pm 51$ | $4567 \pm 109$ | $4848 \pm 81$ | $5170 \pm 68$ | $\mathbf{5229 \pm 36}$ | $4823 \pm 528$ |
| **HalfChee.** | $\mathbf{9347 \pm 206}$ | $9216 \pm 129$ | $8847 \pm 412$ | $8260 \pm 379$ | - | - | - |

blue term in Alg. 2 is biased. We will test two variants, one that ignores the bias and one that corrects it, labeled **DouBel (NB)**. We also test another baseline that include auxiliary losses modeling the environment, but to not clutter further the plots, we defer this comparison to the appendix. To test whether a faster update rate of the target can speed-up SAC, we change the default behavior of doing a Polyak averaging of past networks (Haarnoja et al., 2018) with swapping the target network with the current network every 10 gradient steps. We use this setting for DouBel as well as $\lambda = 0.1$ for all experiments. A second hyper-parameter we experiment with is adding a third hidden layer for the critic network to test whether the faster update rate remains robust to more complex function approximators. As baseline, we use i) SAC with the optimized Mujoco hyper-parameters from `stable-baseline3`, ii) SAC with a similar target update rate as DouBel, labeled **SAC (T10)** and finally iii) SAC with a 3 hidden layer critic labelled **SAC (H3)**. Results are averaged over **20 seeds** averaging the mean episodic return and computing its $95\%$ CI. Table 2 shows a synthesis of all experiments, Fig. 3 shows the learning progress for the three layer comparisons while the comparison against SAC (T10) is deferred to the appendix.

From Table 2, we can see that DouBel (NB) outperforms SAC in 4 environments out of 5 with the biggest difference being on `Walker2d`. Compared to SAC (T10), we see that without the additional BE term, the faster update rate leads to catastrophic consequence on several environments. Fig. 4 shows the Bellman error on these environments and, unsurprisingly, the faster update rate caused a blow up of the error, whereas it remains a lot more contained for DouBel. As for the bias correction of DouBel, there does not seem to be major differences between correcting the bias or ignoring it on these (deterministic) environments. When increasing the number of hidden layers of the critic, DouBel remains stable and in fact widens the gap with vanilla SAC. We also observe improvements for SAC (H3) but the reward curve in `Walker2d` shows some oscillations. Upon closer inspection, we see in Fig. 5 that there are a few seeds that exhibit an erratic behavior whereas all 20 runs of DouBel re-

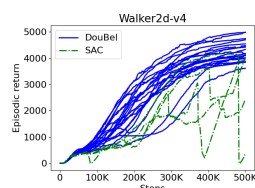

Figure 5: The episodic return of a few unstable runs from SAC (H3) and all 20 runs of DouBel (H3).

main stable. In summary, we have shown that by the simple addition of a Bellman error term to the loss of the critic, the FQE procedure of SAC becomes robust to a wider range of hyper-parameters, including using more aggressive target network update rates or, more surprisingly, deeper critics which all have tangible effects on performance. These results bode well for the wider applicability of our loss to the policy evaluation procedure of other deep RL algorithms.

## 7 CONCLUSION

In this paper we have introduced the DouBel loss that adds an auxiliary squared Bellman error term to the standard fitted Q evaluation procedure. Importantly, only the gradient of the feature function—for example the hidden layers of a neural network—flows through this additional term. This forces the features to be "self-predictive"—without explicitly learning a model—which helps contain the Bellman error and make the fitted Q evaluation more robust to hyper-parameters such as the target network update rate or the number of hidden layers. When integrated into SAC, a state-of-the-art deep RL algorithm, both these hyper-parameters had tangible effects on performance. For future work, one could study the combination of DouBel with variants of SAC that use a larger ensemble of critics or a higher update-to-sample ratio. More broadly, it would be interesting to integrate DouBel into the policy evaluation of other deep RL approaches such as DQN. Finally, finding alternative ways of minimizing $\mathcal{L}$ in a tractable way could be another interesting research direction.

## REPRODUCIBILITY

All our results are averaged over 20 independent runs and we provide each time the confidence intervals of these averages. Our integration of DouBel is made on a widely tested version of SAC with optimized and publicly available hyper-parameters. The datasets used in the policy evaluation section will be made publicly available in addition to the code for both the policy evaluation and the integration to SAC.

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

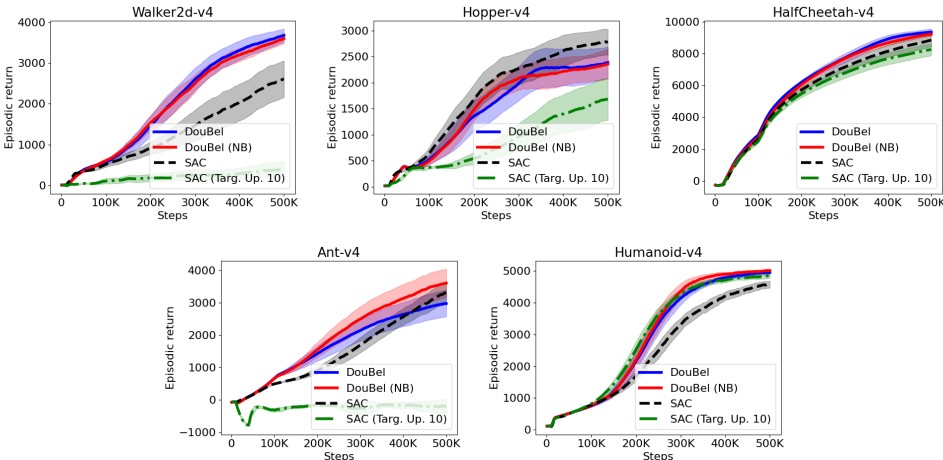

Figure 6: Episodic return for DouBel, SAC and SAC (T10) with two hidden layers. Results averaged over 20 runs showing mean and $95\%$ CI.

## A   APPENDIX

We include in this appendix additional results for which space lacked in the main paper. Namely, results on the two hidden layer setting of SAC, an additional baseline for the experiment in Sec. 6.2, a sensitivity analysis on the parameter $\lambda$, additional experiments integrating DouBel to DQN on five discrete action tasks and finally all the learning curve for the experiments in Sec. 6.1.

### A.1   SAC ON MUJOCO TASKS EXPERIMENTS

**Execution times.**   In our experiments, the execution time of DouBel was about $\sim 1.2$ times the execution time of vanilla SAC on all Mujoco tasks, while the execution time of DouBel with bias correction was $\sim 1.5$ times the execution time of vanilla SAC. This is due to the additional V-value network for the bias correction. Note that to keep comparisons fair, we did not use the value network beyond the bias correction, but in practice the time overhead of learning a value function could be compensated by using the value function to, e.g., reduce the variance of the next state value estimate.

Fig. 6 shows the learning curves of experiments in Sec. 6.2 for SAC with a target update of 10 and no BE auxiliary loss. It can be seen that on some tasks, the agent fails to learn completely, which as has been shown in Sec. 6.2, correlates with an explosion of the Bellman error (Fig. 4).

### A.2   COMPARISONS WITH ENVIRONMENT MODELLING AUXILIARY LOSSES

In this section we assess whether adding an auxiliary loss to learn models of the reward and transition function can help stabilize FQE in SAC when the target update rate is set to 10, in the same way that DouBel did. To do so, we add to the FQE loss of SAC an auxiliary loss for predicting the reward and next state features following Chang et al. (2022). We label this baseline `SAC + BC` in Fig. 7. We run again this baseline with 20 seeds and evaluate it for two values of a parameter $\lambda$, that for this baseline, trades-off between the FQE loss and the modeling losses. We see in Fig. 7 that on some tasks, the higher value of $\lambda$ performs better and on some other tasks it perform worse, but for both, the environment modeling auxiliary losses do not stabilize FQE at such a high target network update rate the same way DouBel does.

### A.3   SENSITIVITY ANALYSIS OVER $\lambda$

DouBel critically depends on a parameter $\lambda$ that trades-off between the FQE loss and the BE loss. In all the SAC experiments we have set $\lambda = 0.1$ and in this section we investigate the behavior of DouBel for various values of $\lambda$ both larger and smaller than $0.1$. The experimental setting is that of Sec. 6.2, using a three hidden layer critic. We run again twenty seeds for each value of $\lambda$.

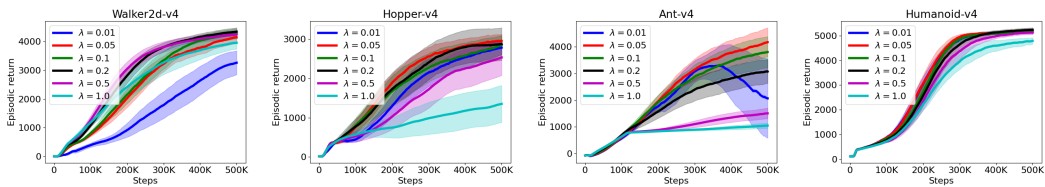

Figure 7: Episodic return for DouBel, SAC and SAC + BC. Results averaged over 20 runs showing mean and 95% CI.

Figure 8: Sensitivity analysis of DouBel to the hyper-parameter $\lambda$. Results averaged over 20 runs showing mean and 95% CI.

Fig. 8 shows that values of $0.1$, $0.05$ and $0.2$ perform rather similarly, with the value of $0.05$ slightly outperforming the value of $\lambda = 0.1$ we used in Sec. 6.2. For the lowest value of $0.01$, SAC shows signs of instability which is not surprising since as $\lambda$ goes to zero we revert to vanilla SAC which we know is unstable for such high update rates of the target network.

More interestingly, when $\lambda$ is too high (values of $0.5$ and $1.0$), we do not observe sharp decreases in policy return in the same way than when $\lambda = 0.01$ on `Ant-v4` but rather that learning becomes slow. A possible explanation could be given by the same mechanisms governing the two-timescale network of Chung et al. (2018). Recall that in the loss $L$ of Sec. 3, we want the weight $w$ to be the LSTD solution of the current feature set. As such, the FQE loss should be optimzed at a faster time-scale than the BE loss following Chung et al. (2018). One way of achieving this is to weight the FQE loss more than the BE loss, which is not the case anymore when $\lambda$ is large.

**Insights on how to set $\lambda$.** In the experiment of Sec. 6.2, since SAC with target update rate of 10 would diverge rather fast, setting an appropriate value of $\lambda$ was relatively easy and consisted in finding the smallest value of $\lambda$ that would stabilize FQE in the first few iterations. We note in addition that both the FQE loss and the auxiliary BE loss have the same scale, so the value of $\lambda$ should be independent of the scale of the reward function or of the current Q-function. $\lambda = 0.1$ could thus be a reasonable starting point for many different tasks.

## A.4 DouBel with DQN.

In this set of experiments, we integrate the DouBel loss into the FQE procedure of DQN on several MinAtar environments (Young & Tian, 2019) and the `LunarLander-v2` environment. The experimental protocol is similar to the integration with SAC, and we set an aggressive target update rate of 1 and see if the additional BE loss can improve over the vanilla algorithm. We note that in these simpler environments, FQE did not diverge as in Mujoco tasks even with such a high target update rate. Fig. 9 shows the performance of DQN and DouBel for various choices of $\lambda$. In these environments and with this target update rate, the value of $\lambda = 0.2$ seems to perform best on all the tasks compared to $\lambda = 0.1$ and $\lambda = 0.05$, and we observe on several games improvements of DouBel over DQN, confirming on a different setting than the SAC + Mujoco tasks of Sec. 6.2 that the added BE loss can improve over vanilla FQE in high target update rate regimes.

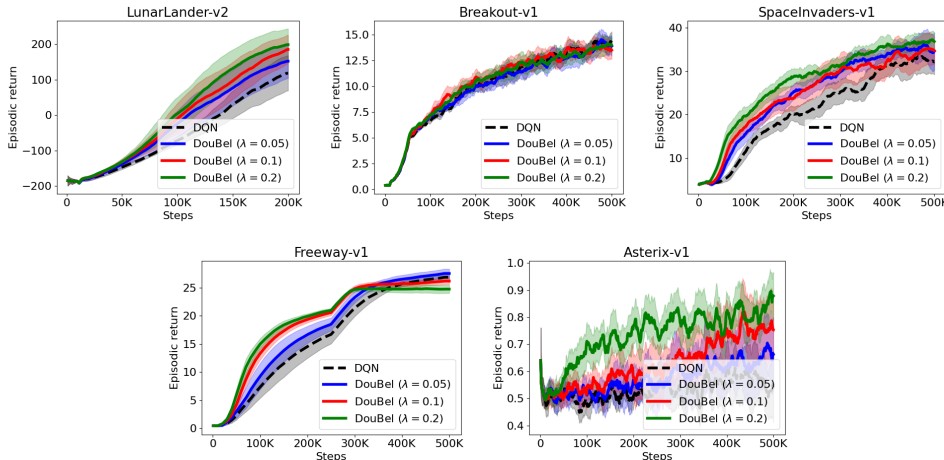

Figure 9: Episodic return of DQN and DouBel for various values of $\lambda$ on finite action space tasks. Results averaged over 20 runs showing mean and 95% CI.

## A.5 HYPER-PARAMETERS

| Parameter | Value |
|---|---|
| $\gamma$ | 0.99 |
| Critic network | (256, 256) |
| Actor network | (256, 256) |
| Activation | ReLU |
| Learning rate | 3e-4 |
| Optimizer | Adam |
| Replay buffer size | 10e+6 |
| Number of burn-in time-steps | 10e+4 |
| $\tau$ | 0.005 |
| Entropy lower bound | MBPO values Janner et al. (2019) |
| Gradients per time-step | 1 |
| Number of critics | 2 |

Table 3: Hyper-parameters of vanilla SAC with a two hidden layer critic network.

| Parameter | Value |
|---|---|
| $\gamma$ | 0.99 |
| Critic network | (256, 256, 256) |
| Activation | ReLU |
| Learning rate | 3e-4 |
| Optimizer | Adam |
| Replay buffer size | 10e+6 |
| Number of burn-in time-steps | 10e+4 |
| Gradients per time-step | 1 |

Table 4: Hyper-parameters for the DQN experiments.

## A.6 LEARNING CURVES FOR THE POLICY EVALUATION SECTION

In this final set of figures, we plot the learning curve (both Bellman error and distance to true Q-function) for the 12 considered datasets.

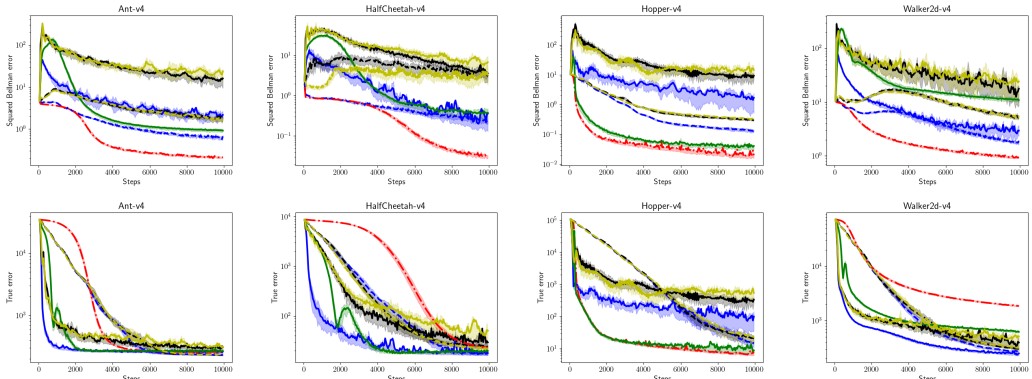

Figure 10: Bellman error (top) and corresponding distance to the true Q function on 4 datasets (`id = 0`), showing the mean as well as the 25% and 75% quantiles. Results averaged over 20 runs.

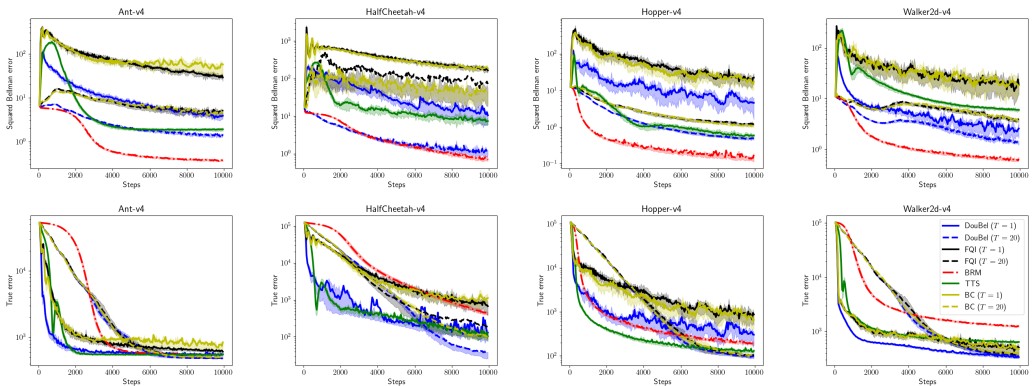

Figure 11: Bellman error (top) and corresponding distance to the true Q function on 4 datasets (`id = 1`), showing the mean as well as the 25% and 75% quantiles. Results averaged over 20 runs

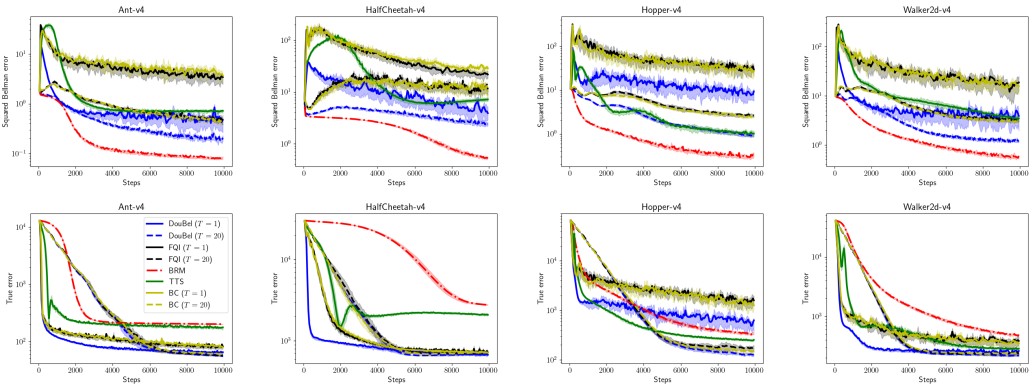

Figure 12: Bellman error (top) and corresponding distance to the true Q function on 4 datasets (`id = 2`), showing the mean as well as the 25% and 75% quantiles. Results averaged over 20 runs

