# OpenReview forum: "Why not both? Combining Bellman losses in deep reinforcement learning"
_ICLR.cc/2024/Conference — Submitted to ICLR 2024_

### Official Review · Reviewer_X6y6 · 2023-10-27

**Soundness:** 3 good
**Presentation:** 2 fair
**Contribution:** 2 fair
**Rating:** 5
**Confidence:** 3

**Summary:**

Some reinforcement learning algorithms usually use a variant of fitted Q-evaluation for policy evaluation, alternating between estimating and regressing a target value function. Based on the prior work, it seems that the Bellman residual can incur poor performance in the linear case or with the neural networks. This paper uncovers that the Bellman residual can be utilized as a useful auxiliary loss for neural fitted Q-evaluation. The authors experimentally show that adding a Bellman residual loss stabilizes policy evaluation. The authors combine the Bellman residual loss with the SAC algorithm, and observe an improved sample efficiency on some tasks while FQE can diverge without the Bellman residual loss.

**Strengths:**

# Strengths

- this paper is highly related to the RL community, especially focusing on the fitted Q evaluation problem

- this paper is generally well-written and well-motivated

- this paper unpacks an interesting conclusion, that the Bellman residual loss can serve as a quite good auxiliary loss for the benefit of improving the sample efficiency. The authors show that utilizing the combination of the projected Bellman error and the Bellman residual can be a better choice.

-  codes are included

**Weaknesses:**

# Weaknesses

- The authors ought to present the theoretical analysis more formally and organize them into theorems like many FQE papers do

- The authors ought to list a detailed hyperparameter setup table in the appendix for clarity

- The hyperparameter $\lambda$ seems to be important and the most critical part of the proposed method. While I do not see enough discussions on this hyperparameter. The authors use different $\lambda$ for different tasks or when combined with different algorithms. It is important to give practical guidance on how to determine this parameter

- Are there any way of tuning this parameter (i.e., $\lambda$) automatically? How does this hyperparameter affect the performance? How sensitive is the method to this parameter? Have you try some other datasets other than MuJoCo? Can your conclusion still hold?

- The authors only combine their method with SAC, then can your method benefit more advanced algorithms like TQC [1], REDQ [2]?

[1] Controlling overestimation bias with truncated mixture of continuous distributional quantile critics. ICML.

[2] Randomized ensembled double q-learning: Learning fast without a model. ICLR.

- Can your method still work when there are already some regularizations on the critic, e.g. DARC [3]? What trade-off we may need to balance the introduced residual loss part and the existing regularization part? I expect some further discussions on this.

[3] Efficient continuous control with double actors and regularized critics. AAAI.

**Questions:**

Please refer to the weaknesses part

---

> ### Author Response · Authors · 2023-11-19
>
> Thank you for your review.
>
>   > 1) Better present the theoretical result
>
> We have now phrased the result as a proposition in the revised paper. Thank you for your feedback.
>
>   > 2) Hyper-parameter table
>
> We have put a hyper-parameter table in the appendix. Thank you for your feedback.
>
>   > 3) Importance and best practices for hyper-parameter $\lambda$
>
>  We have added a sensitivity analysis on $\lambda$ for the SAC + Mujoco experiment and we describe how we ended up choosing $\lambda = 0.1$ for that particular experiment which hopefully provides some helpful insight.  Because FQE diverges relatively quickly in that setting with high update rates of the target, it ended up relatively easy to set a sensible parameter for that experiment. We discuss this in more detail in this new section of the  appendix.
>
>   > 4) Other datasets than Mujoco
>
> We have launched some additional experiments on LunardLander-v2 and MinAtar tasks on DQN. Please refer to the new appendix for more details. These preliminary results indicate that the BE term can also be efficient in stabilizing DQN in high target network update regimes.
>
>   > 5) Integration with other techniques such as ensembling, higher update-to-data ratio, double actors, etc.
>
> There has been a large amount of prior work aimed at improving the sample efficiency of deep RL algorithms. While it is difficult to make definite statements about how this particular auxiliary loss would combine with existing ones, we would like to point out that the mechanisms by which sample efficiency in our case is improved is very unique. Indeed, to the best of our knowledge this regime of high update rate of the target network remains rarely studied as a way of speeding up deep RL. We have shown that it can be impactful on Mujoco tasks and with the new baseline added in the revision, that other modeling-based auxiliary losses are not as effective as the proposed BE loss in preventing divergence of FQE in this regime. Because the properties of our approach are rather unique, we believe there is a good chance that it is complementary with existing methods. Of course successfully integrating these different methods is not trivial but can be an interesting future direction.

---

### Official Review · Reviewer_JxN6 · 2023-10-31

**Soundness:** 3 good
**Presentation:** 2 fair
**Contribution:** 3 good
**Rating:** 5
**Confidence:** 3

**Summary:**

The paper aims to use residual algorithms, which have some nicer theoretic properties but have often performed worse in practice, especially with deep networks. The proposed resolution is derived as follows.

We can first decompose a deep NN $Q(s,a)$ into $\phi \cdot w$ where $\phi$ is a feature vector and $w$ are the linear weights. The w can be viewed as the final layer of an NN which is usually linear. Assume a discrete state and action space, the $\phi$ can define a $\Phi$ "feature matrix" of size $|S x A| x K$ where $K$ is the dimensionality of the feature layer. This is mostly for notation purposes and such a $\Phi$ is never instantiated in practice except for toy problems.

The projected bellman error $||Q - T^\pi Q||$ can be rearranged $||\Phi w - T^\pi Q||$. Following prior work, we can demonstrate that this error can be upper bounded by a model learning loss $L(\phi)$, where $\phi$ are the latent features of the model, and the loss is defined by how well $\phi$ can model the reward function / features of the next state assuming an optimal $w$ for said $\phi$. If I understand the argument correctly, this has not gotten to the new proposed part of this work and is summarizing prior work. The key point is that it shows adding auxiliary model-based losses to a model-free RL algorithm is theoretically justified (and that such losses would be applied to the feature space $\phi$ and ignore updating the final linear weights $w$)

The proposal of this work is to not explicitly use a model-based auxiliary loss. Instead, we should just use something like the Bellman error as an auxiliary loss. We are already using a standard TD-error as our "base" RL loss, but we can take the Bellman residual objective and use this as the aux. loss. Inspired by feature learning work, we only allow this aux. loss to affect the feature layers (every layer except the last one). This still runs into the classic double sampling bias, which we can either choose to correct for or not.

**Strengths:**

The paper provides a helpful primer on Q-learning literature, in particular on the notation norms for considering the $\phi$ vs $w$ decomposition. I appreciate that the plots have error bars and the empirical improvements over baseline SAC seem slightly promising.

**Weaknesses:**

I may be a bit out of the loop on the MuJoCo test suite, but I thought the typical number of environment steps needed per run was on the order of 10^6 steps or higher. But the methods appear to only be benchmarked up to half of that number? This makes me a little suspicious of the results.

From a style standpoint, I'm not sure the theoretical discussion in Section 3 is that helpful and the presentation seems a bit poor. This could be down to me not understanding the paper, but it felt like this:

Figure 1 - a diagram of projection error that has little to do with the proposal to use Bellman residual as an auxiliary loss.

Section 2.1 - A discussion on true bellman error vs projected error, which also has little to do with the bellman residual proposal.

Section 3 - A discussion on feature space learning, where we spend multiple paragraphs and lines of equations deriving losses that show learning good features can improve RL, before saying "but this should be worse than using the Bellman error, let's do something else", which felt a little like it was wasting my time.

The practical implementation of Bellman residual is only brought up around Section 4 and does not seem that related to the sections that come before it.

In terms of experiments, separate from the "seems like too few steps" question, it feels like the paper's argument would be stronger if it included more alternatives for feature learning in the practical section. In Table 1, the BC baseine is good, because it is using a different auxiliary loss than DouBel, and it gives an argument for why DouBel's bellman residual aux. loss is better than a next-state-feature + reward approximation loss. But this table is only for policy evaluation, and then there are no aux. feature methods in Table 2 (episode return) aside from DouBel! So I don't see evidence that DouBel is better than othr auxiliary losses when maximizing episode return, I only see evidence it is better than no aux loss at all.

I think the experiments do support better policy evaluation, but they don't support better policy learning strongly enough. This combined with some of my complaints on presentation make me a bit lukewarm about the work, even if there are some good parts within it.

**Questions:**

I may have missed this - in the feature function gradient, how is $\lambda$ defined? Is this a hyperparameter fixed during learning, or is it something implicitly defined by $||w||$ to match Eqn 13?

---

> ### Author Response · Authors · 2023-11-18
>
> Thank you for your review.
>
>   > 1) $L(\phi)$ follows prior work
>
> There seems to be some confusion on what $L(\phi)$ is and what part of our work is novel. To the best of our knowledge, no other prior work proposes to minimize $L$ for non-linear policy evaluation. If the Reviewer has references showing otherwise, we would be glad to cite them.
>
> We would also like to clarify that $L$ is a Bellman error, where the linear part is the minimizer of the projected Bellman error. $L$ itself is not a model learning loss but can be shown to be bounded by one. To the best of our knowledge this insight is also novel and is the main motivation for our research question: if the usual model learning auxiliary losses are upper bounds to a Bellman error, why not minimize directly the Bellman error as an auxiliary loss?
>
>   > 2) Section 3 unrelated to the rest of the paper
>
> The main point of Section 3 is to show the relationship between existing work on regularising policy evaluation algorithms with auxiliary modeling losses and the proposed Bellman error $L$. The conclusion of Section 3 is not “let us do something else” but instead, why not minimize the Bellman error $L$ directly instead of the upper bounding model learning losses of prior work.
>
> Optimizing $L$ directly is challenging due to the numerical instability of computing the LSTD solution [1]. The practical algorithm is to use instead the current $w$ from an FQE procedure running concurrently as an approximation to the LSTD solution and use this $w$ for computing $L$. While different from computing $L$ exactly, we believe this still lets us investigate the research question raised by Section 3 which is about the merits of using a Bellman error instead of the environment modeling error.
>
> We hope this clarifies better the relation between Section 3 and the remainder of the paper. Otherwise we would really appreciate some additional feedback to improve the presentation of the paper.
>
> Now because $L$ combines both a Bellman error and the minimizer of the projected Bellman error, we believe the refresher in Section 2 is still useful to fix the terminology.
>
> [1] Scherrer; Should one compute the temporal difference fix point or minimize the bellman
> residual? the unified oblique projection view.  ICML10
>
>   > 3) Is $10^6$  time-steps not the norm for MuJoCo?
>
> Prior work already considered shorter total time-steps. E.g. REDQ [CITATION] has Mujoco experiments with 120 to 300K time-steps. In our setting the improvement in sample efficiency is most apparent around the 250K time-step mark and our plots are meant to focus on this region.
>
>   > 4) Lack BC baseline for the SAC + MuJoCo experiment
>
> Thank you for your suggestion. We have now added this baseline.
>
>   > 5) How is $\lambda$ defined
>
> Lambda is a hyper-parameter. We have now included additional plots in the appendix to show the impact of this hyper-parameter.

---

> > ### Comment · Reviewer_JxN6 · 2023-11-22
> >
> > > There seems to be some confusion on what $L(\phi)$ is and what part of our work is novel.
> >
> > Sorry, I do understand that $L(\phi)$ is a Bellman error that is upper bounded by the $\min_{m_r, M_\phi}$ expression that can be interpreted as a model learning loss. I made a mistake when typing up the review and said $L(\phi)$ equaled that upper bound by accident. Thanks for clarifying this.
> >
> > > On Section 3
> >
> > Presentation wise my suggestion would be to do one of the following.
> >
> > 1. Rename the section title from "Combining Bellman Losses" to something more specific to your stated goal of showing the relationship between prior auxiliary modelling losses work and your work.
> > 2. Move the justification / connection to after the practical implementation step, so that the reader has more context on how the bellman losses are combined before reading the justification of why this should be helpful.

---

> > > ### Author Response · Authors · 2023-11-23
> > >
> > > Dear reviewer,
> > >
> > > We have updated the title of Section 3 and added a subsection to state more clearly the goals of this section.
> > >
> > > Regarding the ordering of the sections, Section 3 references a lot of the equations introduced in Section 2 and we think the flow of the paper is more natural if both these sections are adjacent and the practical implementation (Section 4) is closer to the experiments section. For now we have left the order unchanged but we are open to reorganizing the paper if there is a larger consensus among reviewers that swapping the order of what presently are Section 3 and 4 would improve the readability of the paper.
> > >
> > > Thank you for your feedback.

---

### Official Review · Reviewer_BJvN · 2023-11-01

**Soundness:** 3 good
**Presentation:** 4 excellent
**Contribution:** 3 good
**Rating:** 6
**Confidence:** 3

**Summary:**

This paper studies the use of projected Bellman error (PBE) or the mean squared Bellman Error/Bellman residual (BE). the authors show that although BE is not great on its own, it can be a useful auxiliary loss for neural fitted Q-evaluation. Authors provide theoretical results show that existing auxiliary losses that model reward and transition dynamics can be seen as a combination of PBE and BE, and this motivates the design of a new auxiliary loss, the Double Bellman (DouBel) loss. Empirical results are further provided to show that by using the proposed loss on SAC, it is possible to achieve better loss and performance on MuJoCo benchmark, and allow a more frequent target network update.

**Strengths:**

**originality**
- the paper presents a very interesting novel insight making connection between the Bellman losses and the commonly used forward dynamics and reward prediction auxiliary losses.
- empirical results are provided to further support the theoretical results.

**quality**
- the writing and structuring of the paper are good.
- good covering of related works
- the proposed method is well-motivated.
- many technique details are provided, enough seeds are run

**clarity**
- the results and arguments presented in the paper are clear and easy to follow
- figures and tables are clear

**significance**
- the theoretical insight in the paper is quite interesting.
- the empirical results show that the proposed method can indeed achieve better performance and smaller losses. When comparing to SAC, the proposed method is less prone to divergence and can allow faster target network updates. The proposed method can be a nice way to improve algorithm stability. And I believe this applies to not just SAC but other related algorithms as well.

**Weaknesses:**

I think the paper is very interesting but can be nice to see a bit more empirical study and analysis.
- To my understanding, a hyperparameter (lambda in algorithm 2) is used to balance how much auxiliary loss to use, can you provide more ablation on how the algoirthm's behavior and how the accuracy of its Q estimates change as lambda changes?
- In some recent works it has been shown that techniques that provide more accurate Q estimates can be especially helpful when the algorithm is taking more udpates per data point collected. Will the proposed method also benefit from this setting?
- How much computation overhead does the proposed method have? Would like to see a table comparing wall clock time between it and SAC baseline.
- Will the proposed method also lead to better long-term performance?

**Questions:**

- Given the same computation budget, will the proposed method be more efficient compared to methods with other auxiliary losses or with ensemble-based bias reduction?
- Will it be beneficial to combine the proposed method with other auxiliary losses or other bias reduction techniques, or that does not make sense?

---

> ### Author Response · Authors · 2023-11-18
>
> Thank you for your review
>
>   > 1) Sensitivity to hyperparameter $\lambda$
>
> Thank you for your feedback. We have run additional experiments on the SAC + Mujoco setting with 4 additional values of $\lambda$ on the previous 4 Mujoco tasks and averaged over 20 seeds. Results and discussion can be found in the appendix. Please let us know if this addresses your concerns.
>
>   > 2) Will the method benefit from more gradient updates
>
> The way this paper aims at improving sample efficiency is by recomputing the target more often. This leads to more frequent applications of the Bellman operator and lets the rewards propagate faster through the chain of state-action pairs. This shows well in the policy evaluation experiment where we can compute the distance to the true Q function and has tangible benefit when solving RL tasks as shown in the SAC + Mujoco experiments. To the best of our knowledge, no other work has investigated improving sample efficiency by making FQE more robust to faster target value recomputations. We think because its properties are unique, it is more likely that our work can integrate well with prior improvements of SAC/deep RL such as increasing the Q-function ensemble size or performing more gradient steps.
>
>   > 3) Computation overhead
>
> In the SAC + Mujoco experiment, the execution time of DouBel was about ~1.2 times  the execution time of vanilla SAC, while DouBel with bias correction was ~1.5 times the execution time of vanilla SAC, because of the extra V-value network. We have added this to the appendix, thank you for pointing it out.
>
>   > 4) Will the proposed method lead to better long-term performance
>
> We have not observed better long-term performance in the Mujoco task. We think the main benefits of our methods are in the initial speed-up provided by a more frequent application of the Bellman operator. That is why our results focus on the 0-500K range for the SAC + Mujoco task.
>
>   > 5) Efficiency compared to other approaches and possible combinations
>
> Methods such as AQE can be about 200 times slower than SAC, because it has a 10 times larger ensemble size and uses 20 times more gradient steps per sample. Under a fixed computational budget this might not even be more efficient than vanilla SAC. Of course this all depends on the environment and if the cost of interaction is very high (e.g. robotics) it might still be beneficial to focus on sample efficiency first and foremost. As we have shown in the new baseline requested by Reviewer JxN6, auxiliary losses such as those predicting next state features do not stabilize FQE in the high update rate of the target regime, and hence the advantages of our methods are unique. Because of that we believe it can complement quite well prior work towards the goal of developing increasingly more sample efficient deep RL algorithms. Of course merging all these approaches is not necessarily trivial but can be an interesting future direction.

---

### Official Review · Reviewer_he6H · 2023-11-03

**Soundness:** 3 good
**Presentation:** 2 fair
**Contribution:** 2 fair
**Rating:** 5
**Confidence:** 3

**Summary:**

The paper proposes using the Bellman Error (BE) as an auxiliary loss in combination with the Projected Bellman Error (PBE) for Fitted Q-Evaluation (FQE). When the action-value function $Q(s,a) = \Phi w$ is expressed as the product of a learnable feature vector $\Phi$ and a learnable weight $w$, we can find the parameters $w$ minimizing the PBE in closed form by following the standard LSTD solution. Meanwhile, the feature vector $\Phi$ can be minimized using the BE. The authors suggest an adaptation of FQE that iteratively minimizes these two objectives. They provide an upper bound of the BE loss when $w$ is the solution of the PBE, which depends on the reward function and the next state-action feature. For practical implementation, they propose a model-free algorithm that does not require estimating these quantities. Experimental results demonstrate that the addition of a BE auxiliary loss makes the Soft Actor-Critic algorithm more stable, especially when increasing the number of gradient steps before updating the target network, resulting in improved sample efficiency.

**Strengths:**

- The paper offers a comprehensive and clear presentation of the differences and relationship between the Bellman Error and the Projected Bellman Error.

- To my knowledge, the application of the BE as an auxiliary loss alongside the PBE loss is a novel approach.

- The experiments show a decrease in distance to the true Q function for the proposed method and an increase in sample efficiency, which shows that the proposed auxiliary loss can be effective in practice

**Weaknesses:**

- The presentation related to the theoretical results is, in general, clear, apart from equation 13: what are $m_r$ and $M_{\Phi}$?

- In the paper, it is claimed that the proposed method allows for more aggressive target network update rates. However, from the text, I could not understand what exactly that means: is it the case for DouBel(20) or DouBel(1). This generated some confusion throughout the text. I suggest to add an explanation in the text on how target networks are used and why they are important in this setting.


- The paper lacks a simple experiment (e.g., with finite state and action spaces) where the theoretical results can be shown to hold true without approximating the solution of LSTD. It would be beneficial to demonstrate how the auxiliary loss aids such settings before introducing approximations.

- From Table 1, it seems that both DouBel(20) and FQE(20) have much lower loss than DouBel(1) and FQE(1). I would expect to see for some of the datasets presented a plot showing the final distance to the true Q function as a function of the number of gradient steps before the target network is updated (e.g., from 1 to 40). Is the loss in DouBel always lower than the loss in FQE, or is there a trade-off?

- It is not clear to me how the theoretical results imply that the algorithm can have more aggressive target network update rates. I understand that the use of the auxiliary loss can decrease the error in the Q function. However, could the author clarify why this matters when having more target network updates?

**Questions:**

I incorporated most of the questions above. Other minor questions:

- What is the difference between Figures 6, 7, and 8?

- Why is there an initial divergence in Figure 4 for SAC, and how does it relate to the theoretical results?

---

> ### Author Response · Authors · 2023-11-18
>
> Thank you for your review.
>
>   > 1) What are $m_r$ and $M_\Phi$
>
> $m_r$ and $M_\Phi$ define linear models in $Phi$ that minimize the squared error to the reward $\cal R$ and the next state features $\Phi’$.  We have expanded the explanation in page 5. We hope it is clearer now.
>
>   > 2) Clarifying what are (aggressive) target network update rates
>
> We have updated Sec. 2.2 to clarify what target update rates mean. Please see the bolded text.
>
> As for what aggressive update rates are, it depends greatly on the task. For example, tuned hyper-parameters in stable-baseline 3 suggest an update rate of 10 for CartPole but 1000 for Atari.
>
> In our SAC + Mujoco experiment, we consider the update rate of 10 as aggressive since vanilla SAC, without the proposed regularizer, ‘diverges’ in many cases.
>
> For the policy evaluation task, at the fastest update rate of 1, FQE(1) does not diverge but adding our regularizer has a big impact on the distance to the true Q function, whereas for an update rate of 20, DouBel(20) and FQE(20) are about the same in performance and convergence profile.
>
>   > 3) Adding a simple experiment
>
> We experimented with small scale tabular environments with rich observations (i.e. the MDP is finite and small but the learner observes high dimensional vector valued states).  However, in these simple MDPs, Neural FQE was converging very fast to the true Q function even with a target update rate of 1. These simple MDPs were thus not able to illustrate the challenges of policy evaluation with neural function approximators that one can experience on more complex MDPs.
>
>   > 4) DouBel(20) and FQE(20) have much lower loss than DouBel(1) and FQE(1)
>
> Yes, experiments in 6.1 show that if the computational budget allows it, the slower update rate of 20 is better. However, the learning curve illustrates well how slow policy iteration can be in this regime. Because the targets are updated less frequently, the Bellman operator is applied less often and it takes more time to reach a fixed point. To give an idea of the scale of the X axis for the policy evaluation experiments: we perform up to 10K gradient steps, but an algorithm such as PPO on Mujoco performs typically only 320 gradient steps per iteration for its policy evaluation! Even after 10K gradient steps, there are still four Mujoco datasets where DouBel(1) outperforms DouBel(20). Given how computationally expensive deep RL can be, we believe it is still important to consider improving neural FQE through this faster update of the targets. One can always revert to slower update rates once the error reduces enough. The claim of the paper still holds that these faster update rate regimes do not seem helpful with vanilla FQE but become interesting when equipped with our proposed regularizer.
>
>   > 5) DouBel always lower than FQE
>
> In the policy evaluation experiment, both algorithms performed about the same for the update rate of 20. For the SAC experiments, we have included a new parameter sensitivity analysis and it shows that a too high $\lambda$ can be detrimental to performance. Some care should definitely be taken in tuning it, although the value of $\lambda=0.1$ seems to work well for Mujoco tasks and also the new DQN experiments we added to the paper.
>
>   > 6) Relation between theoretical results and more aggressive update rates
>
> The theoretical results relate the usual auxiliary losses used in prior work based on modeling the reward and next state features with the proposed loss $L(\Phi)$. It is not about the learning dynamics of neural FQE. A hand-wavy argument could be that Bellman error minimization is stable because it is a true gradient descent method (unlike FQE), and adding an explicit BE term seems to be effective at avoiding gradient updates of the feature function that would blow-up the Bellman error. However, the only validation of this property in the paper is empirical.
>
>   > 7) Difference between Figures 6, 7, and 8
>
> As can be seen in Table 1, the policy evaluation experiments are done on three datasets for each of the four Mujoco tasks corresponding to different performance levels of the data generating policy. Figures 6, 7, 8 show the learning curves on four Mujoco tasks for each of the three performance levels.

---

> > ### Author Response · Authors · 2023-11-18
> >
> > > 8) Initial divergence in Figure 4 for SAC and relation to theoretical results
> >
> > Note that in Figure 4, it is not SAC with default hyper-parameters but with a more aggressive target network update rate every 10 gradient steps. We doubt there are theoretical results that would explain why it happens only in the beginning since there are counter examples where it happens later in the learning cycle (e.g. Figure 4, right). If the question was why it happens in general, to the best of our knowledge there are only convergence results for neural FQE with very specific (shallow, 1 hidden layer) neural architectures [1] that do not apply here, and as far as we know there are no theoretical results on why FQE would necessarily converge to begin with.
> >
> > [1] Gaur et al.; On the Global Convergence of Fitted Q-Iteration with Two-layer Neural
> > Network Parametrization; ICML23

---

### Author Response · Authors · 2023-11-11
**Thank you for your feedback**

We would like to thank the reviewers for their very diverse and interesting feedback. A recurring concern seems to be the sensitivity of policy evaluation to the regularizer weight $\lambda$. In addition, reviewer JxN6 would like a baseline with an other auxiliary loss for the SAC experiments and has concerns about the presentation of the paper.

We will do our best to get the requested experimental results as early as possible to advance the discussion, and we thank the reviewers again for their feedback.

---

> ### Author Response · Authors · 2023-11-18
>
> Before answering individually to each reviewer, we would like to recap the changes of the revised submission following reviewers’ feedback.
> 1-  A new sensitivity analysis experiment on the SAC + Mujoco setting has been added to analyze the effect of  $\lambda$ on DouBel, as requested by reviewers BJvN and X6y6.
> 2- A new baseline SAC + BC has been added as requested by Reviewer JxN6
> 3- Experiments on different environments as requested by Reviewer X6y6 using this time DQN as a base algorithm.
>
> We have also made several adjustments to the presentation following the feedback of the reviewers. We believe that the incorporation of the reviewers’ feedback has significantly improved the paper and we thank them again for their careful consideration.

---

### Meta-Review · Area_Chair_7GcB · 2023-12-10

**Metareview:**

The paper focuses on policy evaluation and modified fitted Q evaluation to incorporate Bellman residual error as an auxiliary. The paper argues that doing so is beneficial through several empirical results.

Strengths:

- The perspective of using Bellman residual as an auxiliary loss is novel.

Weaknesses:

- For a theoretically motivated work, this paper lacks clear theoretical statements. Upon request from the reviewers, the authors provided their argument as proposition 3.1. Yet, there is no logical or theoretical argument connecting this proposition and the approach proposed other than phrasing an argument in the form of a question: “can we directly use the Bellman error L as an auxiliary loss instead of its upper bounding model learning terms.”

- From thereon, the paper goes through a series of heuristics and intuitive relaxations to arrive at the algorithm proposed.

- Additionally, as the proposed method is about 1.5 times more expensive than SAC, which itself is quite expensive, a fair experiment should allow SAC to use more computation to achieve parity, such as by using more critic updates.

The lack of precise theoretical claims leaves much to be desired, giving a clear appearance of an incomplete paper at this point.

**Justification For Why Not Higher Score:**

There are sparks of ideas here and there, but the paper lacks precise theoretical claims or insights.

**Justification For Why Not Lower Score:**

N/A

---

### Decision · Program_Chairs · 2024-01-16

Reject